# Disease Activity-Dependent Siglec-1 Expression on Monocyte Subsets of Patients with Idiopathic Inflammatory Myopathies

**DOI:** 10.3390/ijms26104950

**Published:** 2025-05-21

**Authors:** Sándor Baráth, Melinda Nagy-Vincze, Zsuzsanna Kun, Dorottya Szinay, Zoltán Griger, Tibor Gábor Béldi, Katalin Szabó, Marianna Száraz-Széles, Zsuzsanna Hevessy, Zoltán Griger

**Affiliations:** 1Department of Laboratory Medicine, Faculty of Medicine, University of Debrecen, 4032 Debrecen, Hungary; barathsa@gmail.com (S.B.); zsuzsannakun02@gmail.com (Z.K.); szeles.marianna@med.unideb.hu (M.S.-S.); hevessy@med.unideb.hu (Z.H.); 2Division of Clinical Immunology, Department of Internal Medicine, Faculty of Medicine, University of Debrecen, 4032 Debrecen, Hungary; melinda.nagyvincze@gmail.com (M.N.-V.); jemappelledori@gmail.com (D.S.); zgriger2005@gmail.com (Z.G.J.); bldtibor15@gmail.com (T.G.B.); szabo.katalin15@gmail.com (K.S.)

**Keywords:** myositis, idiopathic inflammatory myopathies, dermatomyositis, Siglec-1, type-I interferon, biomarker, disease activity, anifrolumab, flow cytometry

## Abstract

Interferon signature is one of the key pathogenic pathways in idiopathic inflammatory myopathies (IIMs), particularly in dermatomyositis (DM). The aim of this study was to analyze Siglec-1, an interferon-inducible receptor, on different monocyte subsets in IIM subtypes and investigate its association with disease activity measures. Siglec-1 expression was measured by 8-color flow cytometry in 62 IIM patients and 10 healthy controls (HC). Disease activity was assessed using the International Myositis Assessment and Clinical Studies (IMACS) core set measures. Active DM patients exhibited significantly higher Siglec-1 mean fluorescence intensity (MFI) compared to inactive subgroups and HCs in every monocyte subset. Intermediate monocytes displayed the highest Siglec-1 expression across all groups and the strongest associations between disease activity markers. Siglec-1 expression on monocyte subsets was strongly associated with global, extramuscular global, constitutional, cutaneous, muscular, and gastrointestinal activity measures, but not with pulmonary, skeletal, and cardiac activities in the DM population. The best indicator of DM global disease activity among the examined biomarkers was Siglec-1 MFI on intermediate monocytes. Siglec-1 on intermediate monocytes correlates strongly with organ-specific clinical and biochemical markers of disease activity; therefore, it is a candidate biomarker for monitoring IIM disease activity. Siglec-1 could be useful in patient selection for interferon-targeted treatments.

## 1. Introduction

Idiopathic inflammatory myopathies (IIM), collectively known as myositis, are rare, heterogeneous autoimmune disorders leading to symmetric proximal muscle weakness, characteristic skin symptoms, and variable internal organ involvement [1]. The disease can be classified into several clinicopathologically-defined subtypes, such as dermatomyositis (DM), polymyositis (PM), and inclusion body myositis (IBM). Nowadays, PM is a rarely detected entity; increasing knowledge about new myositis-specific autoantibodies has led to even more detailed differentiation, and autoantibody-defined subgroups are characterized by distinct clinical phenotypes and muscle biopsy features such as antisynthetase syndrome (ASyS) and immune-mediated necrotizing myopathy (IMNM) [2]. IIMs are intrinsically linked to immune dysregulation, with a significant overlap of clinical and serological markers shared with other autoimmune diseases, including systemic lupus erythematosus (SLE) and systemic sclerosis. These diseases are often associated with a dysregulated immune response, characterized by the production of autoantibodies targeting nuclear and cytoplasmic components [3].

Studies have shown that type I interferons (IFN) play a critical role in the etiopathogenesis of inflammatory myopathies, especially in dermatomyositis [4,5]. It was shown that IFN-stimulated gene expression correlates with disease activity in DM and Juvenile DM (JDM), suggesting a central role of this cytokine family in promoting inflammatory responses. Elevated levels of type I IFNs in blood and tissue samples from IIM patients reflect their role in sustaining chronic inflammation and tissue damage [6,7,8,9,10,11,12,13,14]. This pathway’s activation is often driven by endogenous danger signals, including nucleic acids released from damaged cells, which activate Toll-like receptors and cytoplasmic sensors to stimulate IFN production.

Targeting the IFN pathway has shown promise in managing IIM. The use of Janus kinase (JAK) inhibitors, such as tofacitinib, has demonstrated efficacy in reducing IFN-mediated inflammation and clinical disease activity in refractory DM [15]. Dazukibart, an IFNβ-specific monoclonal antibody, resulted in a marked reduction in disease activity of myositis patients in a phase 2 trial [16], supporting IFNβ inhibition as a highly promising therapeutic strategy in adults with DM. However, type I interferon activation varies individually and is missing in some patients. Therefore, it would be crucial to administer the appropriate treatment for selected patients.

Siglec-1 (CD169) has emerged as a key biomarker for evaluating type I interferon (IFN) activity and disease activity in idiopathic inflammatory myopathies (IIM). Its expression on monocytes correlates with clinical disease activity, response to treatment, and IFN-driven pathology across various subtypes of IIM, particularly in DM. Siglec-1 expression on monocytes is significantly increased in patients with JDM and correlates with the IFN signature [17]. It seems to be a reliable predictor of disease activity and response to treatment, and to be superior to the IFN score in predicting treatment response [17]. In adult IIM, Siglec-1 levels are elevated across subtypes, particularly in DM. Expression correlates with disease activity scores such as the Physician Global Activity (PhGA) and Manual Muscle Testing (MMT). Importantly, Siglec-1 declines with effective treatment, supporting its use in monitoring treatment response [18]. It effectively distinguishes between active and inactive disease and shows potential in predicting treatment outcomes. Additionally, it correlates with histological markers of IFN activity [19]. While the role of Siglec-1 is well-supported in DM, its utility in other subtypes such as immune-mediated necrotizing myopathy (IMNM) and inclusion body myositis (IBM) is less clear, or is controversial [18]. Further research in diverse cohorts is needed to validate its broad applicability.

Siglec-1 (CD169) is an inducible surface adhesion molecule on human myeloid cells, such as monocytes. CD169 seems to crosslink innate and adaptive immunity by antigen presentation and consecutive pathogen elimination, embodying a substantial pillar of immunoregulation [20]. The three major populations of human monocytes are the classical (CD14^+^CD16^−^), non-classical (CD14^dim^CD16^+^), and intermediate (CD14^+^CD16^+^) cell subsets. Classical monocytes have a more pro-inflammatory character, intermediate monocytes are specialized in antigen presentation and play an important role in different infections, while non-classical monocytes present antigen processing capabilities and are responsible for the anti-viral responses of this lineage; they are distinguished from classical monocytes by their association with wound healing processes [21].

However, there are scarce data in the literature about the Siglec-1 expressions in different monocyte subsets of myositis patients, and there are limited available data about the correlation of Siglec-1 expressions with the different myositis activity measures reflecting different organ manifestations. Therefore, the aims of our study were to (1) analyze Siglec-1 expression on classical (cMO), intermediate (iMO), and non-classical monocytes (ncMO) in IIM subtypes; (2) investigate its association with different clinical activity measures; and, finally, (3) test Siglec-1 expression as a biomarker for selecting treatment-refractory patients for interferon-targeted treatment.

## 2. Results

### 2.1. Demographic and Clinical Characteristics of Study Participants

The study included a total of 62 patients with idiopathic inflammatory myopathies (IIM), comprising 33 patients with dermatomyositis (DM), 17 with antisynthetase syndrome (ASyS), and a combined group of five immune-mediated necrotizing myopathy (IMNM), three polymyositis (PM), and four inclusion body myositis (IBM) patients. Additionally, 10 healthy controls (HC) were included (Table 1). Fifty-one patients (82%) had myositis-specific/or myositis-associated antibodies in the sera, where the most frequent antibodies were anti-Jo-1 (17), anti-TIF1γ (10), and anti-Ro52 (7). The proportion of female participants was 54.8% in the overall IIM group, with similar distributions across subgroups: 51.5% in DM, 64.7% in ASyS, and 50% in the IMNM/PM/IBM group. Among healthy controls, 80.0% were female. The mean age of IIM patients was 55.27 ± 14.5 years, with subgroup averages of 52.97 ± 15.4 years in DM, 55.35 ± 10.1 years in ASyS, and 61.5 ± 16.13 years in the IMNM/PM/IBM group. There was no statistically significant difference in age (*p* = 0.098) and sex (*p* = 0.178) between IIM patients and controls. These demographic characteristics provide context for interpreting subsequent immunological and clinical findings in the study. Demographic characteristics, disease activity scores, and antibody profiles are presented in Table 1.

### 2.2. Siglec-1 Expression Across Monocyte Subsets in IIM Patients and Healthy Controls

We measured Siglec-1 expression levels in patients with idiopathic inflammatory myopathies (IIM), including dermatomyositis (DM), antisynthetase syndrome (ASyS), immune-mediated necrotizing myopathy (IMNM), inclusion body myositis (IBM), and polymyositis (PM). Due to the limited number of cases in the IMNM, IBM, and PM subgroups, these patients were analyzed as a single group in further statistical evaluations. We compared the mean fluorescence intensity (MFI) of Siglec-1 across classical, intermediate, and non-classical monocytes of the patients and in a healthy control group. Siglec-1 MFI was significantly higher on intermediate monocytes compared to other monocyte subsets in the patient group, and it was higher than in non-classical monocytes in healthy controls (Figure 1A). These results indicate that intermediate monocytes consistently exhibit the highest Siglec-1 expression. We could not detect any significant differences in Siglec-1 expression on intermediate monocytes in different myositis subgroups and healthy controls (Figure 1B), but Siglec-1 expression on the different monocyte subsets of myositis patients was very heterogeneous. Some patients expressed high Siglec-1 MFI, while some patients’ expressions were comparable to the control. No significant differences in total monocyte percentages were found between patient groups. Looking at the subgroups, the only difference was found in the ASyS group, where the percentage of classical monocytes was increased. Data for each patient group and healthy controls are shown in Appendix A.

### 2.3. Increased Siglec-1 Expression in Active Dermatomyositis

In our analysis of Siglec-1 expression across different monocyte subsets, we observed the highest expression on intermediate monocytes. To further investigate potential differences related to disease activity, we compared the mean fluorescence intensity (MFI) values of Siglec-1 between active and inactive patients within each disease subgroup (IIM, DM, ASys, IMNM/IBM/PM). Disease activity was classified based on the Physician Global Activity score, with patients scoring 4 or higher considered active. Notably, a significant difference in Siglec-1 expression was found in dermatomyositis (DM) patients, where active disease was associated with higher MFI values (*p* = 0.01). This suggests that Siglec-1 expression on intermediate monocytes may serve as a potential biomarker for disease activity in DM. However, no significant differences were observed in the other disease subgroups (Figure 2).

### 2.4. Correlation of Intermediate Monocyte Siglec-1 Expression with Disease Activity Parameters in DM

To further explore the clinical relevance of Siglec-1 expression on intermediate monocytes (iMO) in dermatomyositis (DM), we analyzed its correlation with various disease activity parameters. A significant correlation was observed between iMO Siglec-1 expression and PhGlobal (r = 0.44), Skin-VAS (r = 0.59), MMT-150 (r = −0.58), Extramuscular-Global VAS (r = 0.52), Constitutional VAS (r = 0.6), Cutaneous Dermatomyositis Disease Area and Severity Index (CDASI) (r = 0.63), Health Assessment Questionnaire (HAQ) (r = 0.51), Gastrointestinal VAS (r = 0.48), and the biomarker LDH (r = 0.61). No significant correlations were detected regarding CK (r = 0.08), CRP (r = 0.08), pulmonary (r = 0.33), skeletal (r = 0.35), and cardiovascular VAS (r = 0.39). These findings suggest that increased Siglec-1 expression on intermediate monocytes may be associated with both muscle and extramuscular disease activity in DM, especially in cutaneous domains. The strength of the correlations, represented by Spearman’s r values, along with their corresponding *p*-values, is detailed in Figure 3. The global disease activity of DM patients significantly correlated with Siglec-1 (r = 0.563; *p* = 0.001) and LDH (r = 0.436; *p* = 0.014), but not with CK (r = 0.269; *p* = 0.143) and CRP (r = 0.119; *p* = 0.523), i.e., the best biomarker reflecting disease activity was Siglec-1. We did not detect significant correlations between activity measures of patients with ASyS and the IMNM/PM/IBM group (Appendix A).

### 2.5. Siglec-1 Expressions with Disease Activity Parameters of Treatment-Refractory Patients Treated with Individualized Medications, Including Anifrolumab

Finally, we wanted to test Siglec-1 as a biomarker of precision medicine, i.e., selecting patients for targeted treatment based on the expression of Siglec-1. First, we consecutively recruited treatment-refractory patients based on the following criteria: (1) PhGA > 4, following at least 3 months of treatment with glucocorticoids ± one immunosuppressant ± IVIG. Five patients were detected (Table 2), two with DM, one with JDM, and two with ASyS. Personalized individual treatments were administered, and disease activity measures were assessed at baseline and three months later. Four out of five patients had elevated Siglec-1 MFI, and one had normal Siglec-1 MFI at baseline. Two DM patients with high Siglec-1 MFI were treated with the selective type-I interferon receptor (IFNAR1) inhibitor anifrolumab, which resulted in a rapid, major improvement based on Total Improvement Score (TIS). One of the ASyS patients with high Siglec-1 MFI was treated with rituximab and IVIG, which was efficacious earlier, and this treatment resulted in a normalization of the CK and a moderate TIS. One DM patient received IVIG treatment, which was efficacious, and an ASyS patient with normal Siglec-1 MFI received rituximab, which resulted in moderate TIS at 3 months.

Figure 4 shows the disease activity measurements, muscle force, and CDASI parameters of a treatment-refractory DM patient (No2). This 45-year-old female patient was diagnosed with anti-TIF1γ and anti-Ku positive dermatomyositis in 2023, affecting the muscle, joints, skin, and esophagus. Repeated cancer screening did not detect underlying malignancy. High-dose steroid methotrexate treatment did not achieve remission; therefore, 2 g/kg IVIG was administered monthly without efficacy. Despite this combined treatment, the patient had very high disease activity (Figure 4B), skin erosions (Figure 4C), muscle weakness, and high (60 mg oral prednisone) steroid demand (Figure 4A). The measurement of Siglec-1 expression of the iMO-s was found to be high (686 MFI, 64%); thus, we decided to add the selective Type-I interferon receptor (IFNAR1) inhibitor anifrolumab. After receiving permission from the national authorities, 300 mg anifrolumab was administered monthly, which resulted in a fast and marked improvement of disease activity. The total improvement score was moderate after one month (52.5) and major after three (72.5) and four (77.5) months. The skin symptoms showed remarkable reduction within three months, and the CDASI score decreased from 64 to 14 in parallel with the Siglec-1 expression of iMOs (from 686 MFI; 64% to 171 MFI; 1.9%). Moreover, the prednisolone dose was tapered from 60 to 10 mg daily, without the relapse of the patient. Similarly, the Siglec-1 expression decreased from 299 MFI; 22% to 89 MFI; 0.3% measured on cMOs and from 762 MFI; 29% to 77 MFI; 0.2% on ncMOs.

## 3. Discussion

In the present study, the expressions of Siglec-1, a candidate biomarker of interferon signature, were assessed on monocyte subsets of myositis patients. We can summarize our recent work as follows: (1) the intermediate monocytes exhibit the highest Siglec-1 expression among monocyte subsets in patients and healthy controls; (2) Siglec-1 expression on iMOs is elevated in active DM and correlates strongly with organ specific clinical and biochemical markers of disease activity; and (3) high Siglec-1 expression on iMOs could be used as surrogate marker of active, interferon driven disease for selecting patients for targeted treatments.

Our data are in line with the results by Graf et al. and Kamperman et al., which show that Siglec-1 can distinguish between active and inactive DM patients [18,19]. Traditional biomarkers of disease activity in patients with IIMs include serum levels of creatine kinase and other muscle-related enzymes, which are part of the IMACS core set measurements. However, the relationship of serum muscle enzyme activity to disease activity is variable, especially in adult and juvenile patients with dermatomyositis and patients with IBM; therefore, a great need exists for the development of more specific and sensitive biomarkers [22]. Based on our results, it was shown that Siglec-1 showed the strongest association with global disease activity of DM patients among biomarkers, such as creatine kinase, LDH, or CRP. Consequently, Siglec-1 expression might better reflect the DM disease activity, especially in those cases where the CK level is not or only moderately elevated. Furthermore, Siglec-1 expression of DM patients was strongly associated with activity of certain organ manifestations, especially the skin, muscle, and gastrointestinal, but not with the skeletal and pulmonary manifestations, which argues for a complex pathomechanism of the different organs. There are other relevant biomarkers, which are related to disease activity, such as cytokines (IL-1β, IL-6, IL-8, IFNα), myeloid cell markers, myositis specific antibodies, or Type I interferon score, which measures the upregulation of type I IFN-stimulated genes (ISGs) in peripheral blood, skin, and muscle [22,23]. Many of them are detected by difficult, time-consuming methods, which limit their feasibility. The detection of Siglec-1 expression on iMOs is easy, fast, and reproducible; therefore, it might be considered to use this biomarker as a part of the disease activity assessment of DM patients.

Precision medicine consists of a tailored approach for each patient, based on individual genetic and immunological variation, which might influence disease course and drug response. Ideally, patients at the time of diagnosis should be subjected to an initial evaluation to profile their disease, with biomarker assessment to identify the main relevant pathophysiological pathways, and prediction of the disease course and stratification of the risk of specific organ damage [24]. Our results showed that the different subtypes of active myositis had significantly different Siglec-1 expressions. The correlation of Siglec-1 expression with disease activity of the ASyS group was not significant; however, some patients had high Siglec-1 expression and high disease activity, while some patients had low Siglec-1 expression and high disease activity, which might reflect different pathogeneses of the disease. In our opinion, Siglec-1 could be used as a surrogate marker of interferon-driven pathogenesis, and measurement of Siglec-1 in treatment-refractory patients could be useful for the decision of treatment selection.

Many clinical trials targeting the interferon pathway are recruiting patients with myositis. Dazukibart (NCT06698796), anifrolumab (NCT06455449), and several JAK inhibitors (NCT04972760; NCT0543726) are currently being investigated [16,25]. Furthermore, recent case reports showed the efficacy of anifrolumab in refractory cases [26,27,28,29]. Based on the improvement of two out of five of our treatment-refractory DM patients treated with anifrolumab, we demonstrate that therapy-refractory dermatomyositis can be effectively treated with targeted interferon inhibition. The efficacy of this treatment is supported by the parallel decrease in disease activity markers and Siglec-1 expression. Based on these findings, measuring Siglec-1 expression could be a valuable tool in selecting appropriate treatment for therapy-refractory myositis patients, as high Siglec-1 levels may predict the benefit and effectiveness of interferon pathway inhibition.

Human monocytes are divided into three major populations: classical (CD14^+^CD16^−^), non-classical (CD14^dim^CD16^+^), and intermediate (CD14^+^CD16^+^). The expression of cell surface markers and their roles in homeostasis and disease set each of these subsets distinct from the others. The ability of classical monocytes to differentiate into monocyte-derived DCs and secrete soluble mediators to bridge innate and adaptive immune responses gives them a more pro-inflammatory nature. Intermediate monocytes are specialized in antigen presentation and play an important role in different infections (Leishmania, HIV). Non-classical monocytes are different from classical monocytes in that they are linked to wound healing processes, and they are responsible for this lineage’s antiviral responses and possess the capacity to digest antigens [21]. Intermediate monocytes express the highest levels of antigen presentation molecules [30,31] and secrete TNF-α, IL-1β, IL-6, and CCL3 upon TLR stimulation [30,32,33]. Szaflarska et al. described an anti-tumoral phenotype for these cells [34]. They express more CCR5 than classical monocytes, contributing to their high susceptibility to HIV-1 infection [35,36,37]. CD14^+^CD16^+^ monocytes expand in systemic infections, indicating a key role in rapid pathogen defense [38,39]. Their increased presence correlates with systemic inflammation in conditions such as coronary artery disease and sepsis [40], suggesting potential as biomarkers for disease severity. In human cardiovascular diseases (CVD) and inflammatory conditions, the inflammatory intermediate CD14^+^CD16^+^ monocyte is increased, and they have been shown to interact with T cells, influencing immune activation and antigen presentation [41].

Thus, there are several arguments in favor of investigating the expression of markers indicating type I interferon pathway activation on intermediate monocytes, such as their role in pathological conditions, expression of activation markers and cytokines, potential role for predicting disease progression, and their role in communication with T cells.

Siglec-1 is a marker associated with monocytes. In our study, its highest expression was observed in intermediate monocytes compared to other subsets. This difference was evident in samples from healthy controls. However, in clinically active patients, Siglec-1 expression was significantly higher than in healthy controls.

Some possible limitations of the study must be mentioned. A key limitation of our study is the small size of active patients and the healthy control cohort, which reduces statistical power and may limit the generalizability of our baseline comparisons. The number of PM, IBM, and IMNM patients was low, and the analysis of these patients in the same group might confound the differences between the subgroups. The analysis of a larger group of patients with IMNM, IBM, and PM subtypes is required to draw conclusions regarding Siglec-1 expression and interferon-driven pathogenesis in these subtypes. Additionally, incomplete data for some clinical and laboratory parameters introduced further uncertainty in our correlation analyses. Finally, although elevated Siglec-1 expression on intermediate monocytes was noted in certain active patients, we also observed unexpectedly low Siglec-1 MFI in a subset of individuals with active disease, suggesting that, in these cases, symptomatology may be driven by mechanisms other than type I interferon-mediated signaling.

## 4. Materials and Methods

In this cross-sectional cohort study, IIM patients under the care of the Department of Clinical Immunology, University of Debrecen, Hungary, were included between June 2024–January 2025. All patients had a definite or probable diagnosis of IIM according to the EULAR/American College of Rheumatology’s (ACR) myositis criteria [42]. This study meets and is in compliance with all ethical standards of medicine. Written informed consent was obtained from all the subjects. This study was carried out in compliance with the Declaration of Helsinki. The study was approved by the Institutional Review Board of the University of Debrecen (Ethical permission number: DE RKEB/IKEB-5404-2020).

A total of 62 patients with idiopathic inflammatory myopathies (IIM) were included in the study, comprising 34 women and 38 men. The average age of the patients was 55.3 ± 14.5 years. The results were compared with those of 10 healthy control individuals. Treatment-refractory patients were followed prospectively, and assessments of disease activity and biomarkers were measured longitudinally.

### 4.1. Evaluation of Myositis Disease Status

Disease activity was evaluated using the International Myositis Assessment and Clinical Studies Group (IMACS) core set measures [43,44]: the physician global activity Visual Analogue Scale (VAS), the Manual Muscle Test (MMT-150), patient global activity VAS, extramuscular global activity VAS, a Health Assessment Questionnaire (HAQ), laboratory values of muscle enzymes (CK—creatine kinase, LDH—lactate dehydrogenase), and myositis disease activity assessment tools (MDAAT) were assessed. Cutaneous activity was determined by the Cutaneous Dermatomyositis Disease Area and Severity Index (CDASI), which is a validated disease severity score of DM [45]. CDASI Activity score ranges 0–100 and Damage score 0–32. Active disease was defined by the physician’s global activity ≥ 4 cm of VAS. Regarding the improvement of the disease activity, the Total Improvement Score (TIS) was used [46].

Laboratory tests included the serum level of creatine kinase (CK), lactate dehydrogenase (LDH), and C-reactive protein (CRP). Immunological analyses included tests for the following autoantibodies: antinuclear antibodies (ANA), anti-centromere antibodies (ACA), anti-histone antibodies, and anti-cytoplasmic antibodies were determined by indirect immunofluorescence on HEp-2 cells (Viro-Immun Labor-Diagnostika GmbH, Oberursel, Germany); ANA positivity was assessed at 1:40 dilution. Titers of the antibodies against extractable nuclear antigen (ENA) complex, anti-SS-A (Ro), anti-SM-RNP, and anti-Jo-1 antibodies were measured (HYCOR Biomedical Inc., Garden Grove, CA, USA) using this latter method. Myositis-specific and associated antibodies (TIF1γ, NXP2, Mi2, SAE, MDA5, Jo1, PL7, SRP, Ku, Pm-scl, Ro52) were detected by membrane-fixed line blots according to the manufacturer’s instructions (Euroline Myositis Antigen Profile4, EuroImmun, Lübeck, Germany). Anti-HMGCR antibodies were detected by indirect immunofluorescence assay and confirmed by in-house ELISA (the cut-off value was ≥40 U/mL).

### 4.2. Flow Cytometry Analysis of Monocyte Subsets

Peripheral blood anticoagulated with EDTA was analyzed by flow cytometry. The cell surface expression of the Siglec-1 (CD169, clone 7-239, Biolegend, San Diego, CA, USA) molecule was analyzed on different monocyte subpopulations using multicolor flow cytometry. The identification of monocyte subpopulations was performed according to literature recommendations using the following marker combinations: anti-CD7-PB (clone 8H8.1, Beckman Coulter, Brea, CA, USA), anti-CD14-FITC (clone MϕP9, Becton Dickinson, Franklin Lakes, NJ, USA), anti-CD16-APC-AF750 (clone 3G8, Beckman Coulter, Brea, CA, USA), anti-CD24-PE (clone ML5, Becton Dickinson, Franklin Lakes, NJ, USA), anti-CD45-PO (clone HI3O, Exbio, Prague, Czech Republic), and anti-CD56-PC7 (clone N901 (NKH-1),Beckman Coulter, Brea, CA, USA). Peripheral blood samples were labeled following the conventional lyse-wash protocol. The measurements were performed on a BD FACSCanto II^TM^ (Franklin Lakes, NJ, USA). A total of 100,000 events were collected per sample. The gating strategy is illustrated in Appendix A. After excluding singlets, live cells were selected based on forward scatter area (FSC-A) and side scatter area (SSC-A) parameters. Monocytes were then gated using a broader gate based on CD45 expression and SSC-A parameters. Subsequently, T cells, NK cells, B cells, and neutrophil granulocytes were excluded from the monocyte gate using the antibodies described in the methods section and were removed through a logical gating approach. The remaining events were considered monocytes, and their subpopulations were defined based on CD14 and CD16 expression [47]. CD169 expression was then analyzed in these subpopulations. Classical monocytes were defined as CD14^+^/CD16^−^, intermediate monocytes as CD14^+^/CD16^+^, and non-classical monocytes as CD14^dim^ or negative/CD16^+^ monocytes. Population identification and capitation were performed on density plots as recommended above [47]. Siglec-1 expression in these populations was characterized by mean fluorescence intensity (MFI) values using the FACSDiva version 6.1.3 software (BD Biosciences, San Jose, CA, USA) statistics module. Granulocyte and lymphocyte populations were used as internal controls as Siglec-1-negative cells.

### 4.3. Statistical Analysis

Statistical analysis was performed using GraphPad PRISM 9.1.2 software. Normality of the data was assessed using the Shapiro-Wilk test. Comparisons and correlations were evaluated based on data distribution (normal or non-normal) using the Mann-Whitney test, Kruskal-Wallis test, and Spearman correlation analysis. Continuous variables were described based on normal distribution by mean and SD values or median and the interquartile range (IQR). Categorical variables were described using frequencies (case numbers) and percentages. A *p*-value of <0.05 was considered statistically significant.

## 5. Conclusions

In conclusion, analysis of Siglec-1 on intermediate monocytes by flow cytometry provides new insights into the pathogenesis of IIM. Siglec-1 on intermediate monocytes is elevated in active DM and correlates strongly with organ-specific clinical and biochemical markers of disease activity; therefore, it is a candidate biomarker for monitoring IIM disease activity. Furthermore, its role in the interferon signaling pathway highlights its relevance in identifying patients who may benefit from interferon-targeted therapies in routine clinical practice.

## Figures and Tables

**Figure 1 ijms-26-04950-f001:**
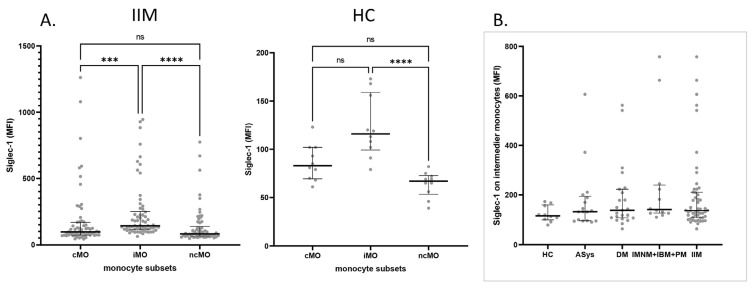
(**A**) Siglec-1 expression on subsets of monocytes in patients with myositis and healthy controls. Regardless of how active the disease was, intermediate monocytes in the overall IIM group (n = 62) had higher Siglec-1 expression than the classical and non-classical subsets. In the healthy control cohort (n = 10), intermediate monocytes exhibited higher Siglec-1 MFI than non-classical monocytes. Although they also trended above classical monocytes, that difference did not reach statistical significance. (**B**) Comparison of Siglec-1 expression on intermediate monocytes of subgroups of patients. Data normality was evaluated by the Shapiro–Wilk test. Group differences were analyzed using the Kruskal–Wallis test and, for individual pairwise contrasts, Dunn’s post-hoc test. Sample numbers in each group: HC n = 10; DM n = 33; ASyS n = 17; IMNM/IBM/PM n = 12. Each dot represents the indicated monocyte subset of a single participant. cMO: classical monocytes; iMO: intermediate monocytes; ncMO: non-classical monocytes; MFI: mean fluorescence intensity; IIM: idiopathic inflammatory myopathies; DM: dermatomyositis; ASyS: antisynthetase syndrome; IMNM: immune-mediated necrotizing myopathy; IBM: inclusion body myositis; PM: polymyositis; HC: healthy control; ns: non-significant; ****: *p* < 0.0001; *** *p* < 0.001.

**Figure 2 ijms-26-04950-f002:**
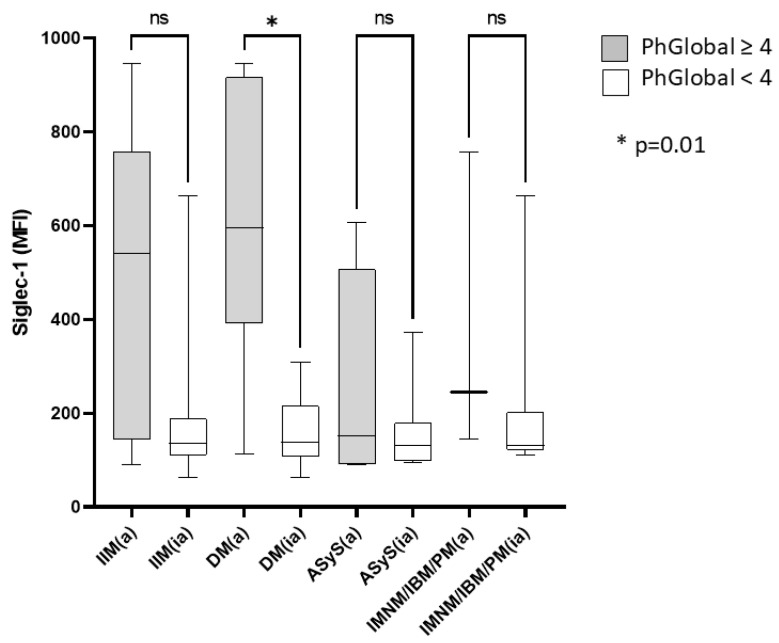
Comparison of Siglec-1 mean fluorescence intensities (MFI) between clinically active and inactive patients. Patients with a Physician Global score equal to or greater than 4 were considered active. Only the DM group showed a statistically significant change in Siglec-1 MFI—note that “IIM” refers to the entire patient population. Data normality was evaluated by the Shapiro–Wilk test. Differences between active and inactive patients were analyzed using the Mann-Whitney test. Sample numbers in each group: DM(a) n = 8, DM(ia) n = 25; ASyS(a) n = 4; ASyS(ia) n = 13; IMNM/IBM/PM(a) n = 3, IMNM/IBM/PM(ia) n = 9. MFI: mean fluorescence intensity; IIM: idiopathic inflammatory myopathies; DM: dermatomyositis; ASyS: antisynthetase syndrome; IMNM: immune-mediated necrotizing myopathy; IBM: inclusion body myositis; PM: polymyositis; ns: non-significant; PhGlobal: Physician Global Activity; a: active; ia: inactive.

**Figure 3 ijms-26-04950-f003:**
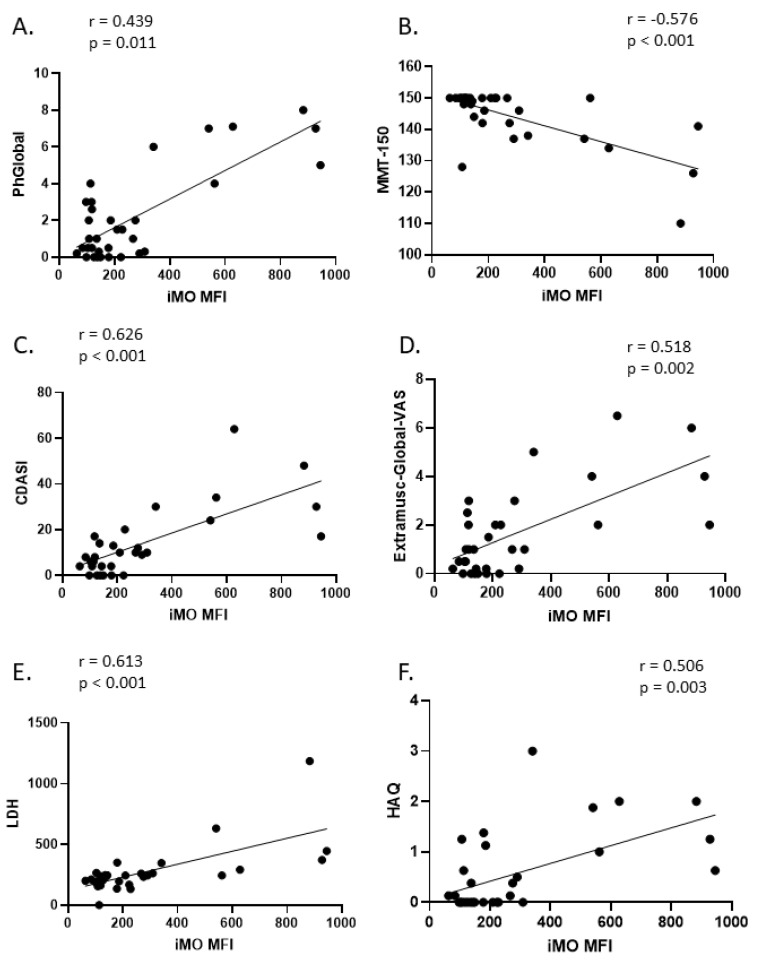
Correlation of Siglec-1 MFI values of intermediate monocytes with disease activity parameters of patients with dermatomyositis. (**A**): Physician Global Disease Activity; (**B**): Muscle force; (**C**): Cutaneous Dermatomyositis Disease Area and Severity Index; (**D**): Extramuscular Global Disease Activity; (**E**): Lactate Dehydrogenase; (**F**): Health Assessment Questionnaire. Data normality was evaluated by the Shapiro–Wilk test. Correlations between Siglec-1 MFI values of intermediate monocytes and disease activity parameters were analyzed by calculating Spearman’s correlation. The sample number in the DM group was 33. Each dot represents the data of a single participant. MFI: mean fluorescence intensity; iMO: intermediate monocyte; PhGlobal: Physician Global Activity; MMT-150: Manual Muscle Testing; CDASI: Cutaneous Dermatomyositis Disease Area and Severity Index; Extramusc-Global-VAS: Extramuscular-Global Activity; LDH: lactate dehydrogenase; HAQ: Health Assessment Questionnaire. Values *p* < 0.05 were considered statistically significant.

**Figure 4 ijms-26-04950-f004:**
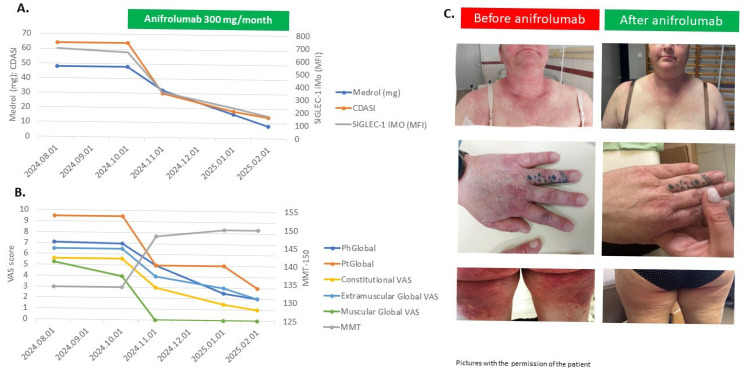
Disease course and Siglec-1 MFI on intermediate monocytes in a refractory anti-TIF1γ DM patient treated with anifrolumab. (**A**) methylprednisolone dosage (Medrol), CDASI values, and Siglec-1 MFI values of intermediate monocytes at baseline and different times of therapy. (**B**) Scores of disease activity measures, including manual muscle force (MMT). (**C**) Skin symptoms before and after anifrolumab treatment. Pictures were taken with the written informed consent from the patient. MFI: mean fluorescence intensity; iMO: intermediate monocyte; CDASI: Cutaneous Dermatomyositis Disease Area and Severity Index; VAS: 10 cm Visual Analog Scale; PhGlobal: Physician Global Activity; MMT-150: Manual Muscle Testing; PtGlobal: Patient Global Activity.

**Table 1 ijms-26-04950-t001:** Main demographic, clinical, and serological characteristics of the study population. Abbreviations: MMT-150: Manual Muscle Testing; IQR: interquartile range; PhGA: Physician Global Activity; PtGA: Patient Global Activity; VAS: 10 cm Visual Analog Scale; CK: creatine kinase; CRP: C-reactive protein; LDH: lactate dehydrogenase; HAQ: Health Assessment Questionnaire; CDASI: Cutaneous Dermatomyositis Disease Area and Severity Index, NA: Not Applicable.

	Total IIM n = 62	DM n = 33	ASyS n = 17	IMNM/PM/IBM n = 5/3/4	HC n = 10
Female, n (%)	34 (54.8)	17 (51.5)	11 (64.7)	6 (50)	8 (80)
Age (mean ± SD)	55.27 ± 14.5	52.97 ± 15.4	55.35 ± 10.1	61.5 ± 16.13	45.9 ± 16.6
MMT-150, median (IQR)	147 (141–150)	150 (141.5–150)	146 (142–149.5)	142.5 (116–148)	NA
PhGA, median (IQR)	2 (0.5–3.57)	1 (0.25–3.5)	2 (1.75–3.58)	3 (1–4.5)	NA
PtGA, median (IQR)	2 (1–5)	1 (0.5–3)	3 (1.5–5)	3 (0.625–5.75)	NA
Extramuscular Global VAS, median (IQR)	1 (0.18–3)	1 (0.2–2.75)	3 (0.5–4)	0 (0–2)	NA
CK (U/L) median (IQR)	116 (66–223)	96.5 (57.3–178)	110 (65–273)	212 (113–723)	NA
CRP (mg/L) median (IQR)	3.1 (1.3–7.9)	2.62 (1.26–7.7)	4.4 (1.4–20.9)	2.73 (1.07–5.19)	NA
LDH U/L median (IQR)	244 (204–278)	236.5 (196–264	252 (230–318)	243 (226–317)	NA
HAQ median (IQR)	0.19 (0–1.25)	0.13 (0–1.07)	0.25 (0–0.63)	1.13 (0–2.32)	NA
CDASI, median (IQR)	4 (0–13.3)	10 (4–17.5)	0 (0–10.5)	0 (0–0)	NA
No antibodies, n (%)	11 (18)	5 (15)	0	6 (50)	NA
anti-TIF1γ	10	10 (30)	0	0	NA
anti-Mi2	4	4 (12)	0	0	NA
anti-SAE	4	4 (12)	0	0	NA
anti-NXP2	2	2 (6)	0	0	NA
anti-MDA5	3	3 (9)	0	0	NA
anti-Jo1	17	1 (3)	16 (94)	0	NA
anti-PL7	1	0	1 (6)	0	NA
anti-SRP	3	0	0	3 (25)	NA
anti-HMGCR	1	0	0	1 (8)	NA
anti-Ku	1	0	0	1 (8)	NA
anti-Pm-scl	3	3 (9)	0	0	NA
anti-Ro52	7	6 (18)	0	1 (8)	NA
anti-SM/RNP	1	0	0	1 (8)	NA

**Table 2 ijms-26-04950-t002:** Data and personalized therapeutic strategies of treatment-refractory patients.

Patient No	1	2	3	4	5
Sex	Male	Female	Male	Male	Male
Age	52	45	24	54	25
Subtype	ASyS	DM	DM	ASyS	JDM
Antibody	Jo1, Ro52	TIF1γ; Ku	Pm-Scl	Jo1	None
Previous Therapy	GC, MMF,	GC, MTX, IVIG	GC, MTX	GC, AZA	GC, MTX
BSL PhGA	4.8	7	7	4,0	7
BSL CK (IU/L)	2496	55	5241	1164	721
BSL Siglec-1 (%)/MFI on iMO	36/606	64/628	25/545	0/100	77/928
BSL GC dose (mg)	60	60	80	10	40
Treatment	RTX, IVIG	Anifrolumab	IVIG	RTX	Anifrolumab
3Mo PhGA	2.5	2	1.5	2.0	2
3Mo CK (IU/L)	144	98	82	242	519
3Mo Siglec-1 (%)/MFI on iMO	NA	1.9/171	NA	NA	94/944
3Mo GC Dose (mg)	15	10	15	5	15
3MoTIS	55	72.5	70	35	75
3MoTIS Category	MODERATE	MAJOR	MAJOR	MINIMAL	MAJOR

## Data Availability

The data that support the findings of this study are available on request from the corresponding author. The data are not publicly available due to privacy or ethical restrictions.

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
