# Peer review of "Disease Activity-Dependent Siglec-1 Expression on Monocyte Subsets of Patients with Idiopathic Inflammatory Myopathies"

_ijms, 2025, doi:10.3390/ijms26104950_

Round 1
Reviewer 1 Report
Comments and Suggestions for Authors
The authors report a screening for siglec-1 expression in different subsets of peripheral blood monocytes in patients with inflammatory myopathies. They aim to find if there is an association with the clinical activity of the disease and if siglec-1 expression could be used as a biomarker for the treatment with type I interferon activity blockers in refractory patients.
The authors study 33 patients with dermatomyositis (DM) but include a few patients of other inflammatory myopathies. For this reason they analyzed polymyositis, inclusion body myositis and immune-mediated necrotizing myopathy as a group. These diseases have very different etiologies and grouping all of them is a serious flaw. Actually, when they analyze the anti-synthetase group, although they have 17 patients there was no correlation between siglec-1 expression and disease activity and they claim that this might reflect different pathogenesis of the disease.
They found that siglec-1 expression was higher in intermediate monocytes both in the patient’s group and in the control group.
They main finding was that siglec-1 expression on iMOs is increased in active DM. However, this was already reported previously in jDM. In fact, the authors state “our data validate the results by Graf et al and Kamperman et al…”
Overall, the study is not completely novel and the analysis of the other groups of inflammatory myopathies requires a larger group of patients before drawing any conclusions.
Author Response
Response to reviewer 1:
First of all, we would like to thank the referee very much for the careful review of our manuscript and the constructive criticism. We vigorously revised the manuscript and made changes according to the comments.
- Comment: The authors study 33 patients with dermatomyositis (DM) but include a few patients of other inflammatory myopathies. For this reason they analyzed polymyositis, inclusion body myositis and immune-mediated necrotizing myopathy as a group. These diseases have very different etiologies and grouping all of them is a serious flaw. Actually, when they analyze the anti-synthetase group, although they have 17 patients there was no correlation between siglec-1 expression and disease activity and they claim that this might reflect different pathogenesis of the disease.
Response: The reviewer has right. Polymyositis, inclusion body myositis and immune mediated necrotizing myopathies have different aetiologies and different, yet not fully understood pathomechanism. Some experts and authors even questioning the existence of polymyositis. However these entities share some similarities too, such as muscle inflammation leading to weakness without characteristic skin symptoms. As our project was a cross sectional study, we included these patients too, but our primary aim was not to analyse IMNM, PM or IBM patients. Unfortunately the lower number of these subgroups led us to group them. The low numbers of the mentioned subgroups did not allow statistical analysis in each subgroup. This is one limitation of our study. Based on the opinion of the reviewers we have highlighted the limitations of the study by inserting a separate paragraph in the discussion section.
- Comment: They main finding was that siglec-1 expression on iMOs is increased in active DM. However, this was already reported previously in jDM. In fact, the authors state “our data validate the results by Graf et al and Kamperman et al…”
Response: We agree with the reviewer that some of our results are in line with previous reports. We rephrased that sentence, which was misleading. However, none of the previous publications studied the Siglec-1 expressions of the different monocyte subsets. The intrinsic heterogeneity of the monocyte compartment prompted us to look deeper. Classical, intermediate and non-classical monocytes carry out distinct functions, so we hypothesized that each subset might differentially up- or down-regulate Siglec-1 in response to type I IFN–driven stimuli. By dissecting these subsets, our goal was to reveal subset-specific patterns that bulk measurements could miss. Furthermore, none of the previous articles analysed the correlation of Siglec-1 expression on different monocyte subsets with organ specific activity measures, such as CDASI, pulmonary, skeletal VAS, etc. Based on the reviewer’s comments we rephrased and included these results in the abstract too.
- Comment: Overall, the study is not completely novel and the analysis of the other groups of inflammatory myopathies requires a larger group of patients before drawing any conclusions.
Response: The reviewer has right, the analysis of a larger group of patients with IMNM, IBM and PM subtypes are required to draw conclusions regarding Siglec-1 expression and interferon driven pathogenesis in these subtypes. However, in this project, we did not aim to analyse these subgroups. The number of our patients (DM: 33, ASyS: 17, IMNM: 5, PM: 3, IBM: 4) are higher than in the article by Kamperman et al (DM: 9, IMNM: 5, ASyS: 1, Overlap: 4), and a little bit less then Graf et al (Graf: DM: 38, ASyS: 19, IMNM: 8, IBM: 9, Overlap myositis: 22). We believe that our manuscript has several novelties, such as (1) the intermediate monocytes exhibit the highest Siglec-1 expression among monocyte subsets in myositis patients and healthy controls; (2) Siglec-1 MFI on iMO-s was strongly associated with global, extramuscular global, constitutional, cutaneous, muscular and gastrointestinal activity measures but not with pulmonary, skeletal and cardiac involvement in the DM population; (3) Siglec-1 MFI on intermediate monocytes was the best indicator of DM global disease activity in comparison with biomarkers, such as CK, LDH and CRP; (4) high Siglec-1 expression on iMOs could be used as surrogate marker of active, interferon driven disease for selecting patients for targeted treatments. Furthermore we presented a case series of treatment refractory patients showing the efficacy of anifrolumab in DM. We fully agree with the reviewer that these novelties were not well presented in the manuscript, therefore we modified the abstract, the result and the discussion sections.
Reviewer 2 Report
Comments and Suggestions for Authors
The article by Baráth et al. reports on the increased expression level of Siglec-1 on intermediate subtype of monocytes from active dermatomyositis patients. The significant correlation between Siglec-1expression and the disease activity is highly relevant and may provide an advanced assessment tool for the diagnosis and treatment of dermatomyositis. The authors have applied state-of-the-art methods and presented the results in a brief and concise manner. The manuscript is well written. Nevertheless, the Abstract, Introduction, Results, and Discussion sections require further revision to improve the overall quality and effectiveness of the manuscript.
Abstract
Line 14: Please provide the number of IIM patients and controls: … in 62 IIM patients and 10 healthy controls (HC).
Line 15: Please provide “the International Myositis Assessment and Clinical Studies (IMACS)” and remove “62 IIM patients participated in the study.”
Lines 19-22: Please rephrase the complex sentence.
Introduction
Line 64-66: Please provide references.
Results
Line 108: Please accurately detail the differences between the control and patient groups. According to Table 1, there are obvious differences regarding age and sex. Other characteristics of the control group are not provided in Table 1(NA). Please address the missing data as they may be relevant in terms of inflammatory status. Which other characteristics have been applied to define the control cases?
Line 123: Figure 1 A shows that only 15-20 patients with IIM express higher Siglec-1 on iMOs. Most of the patients show a lower expression than 200 MFI which is comparable to some controls. In healthy controls, iMOs express significantly higher level of Siglec-1 than noMOs. Please report the results in detail as they appear in the figures.
Lines 131-132: Please explain the diagrams in detail. Figure legends should explain all indicated characters, cMO, iMO, noMO, etc. In figure 1A, MFI is indicated at the x-axis and should be removed. Please indicate that each gray dot represents the indicated monocyte subset of a single participant.
Lines 146-147: Please include all abbreviations, applied statistical tests including normality test, and the number of patients in each subgroup into the figure legend. The figure legends should independently provide the reader a sufficient level of information to understand the presented data.
Line 147: Please revise “. .”.
Lines 167-168: Please follow the comments to figure legends above (Lines 146-147).
Lines 195-196 and 202-203: Please provide the Siglec-1 level of cMOs and nomos before and after treatment.
Figure 4C: The images before and after treatment were obtained from different hands and by different light intensities. Please include better comparable images and crop the facial area as they are not relevant. I suggest the attached layout.
Discussion
The authors are kindly advised to reveal the limitations of their study, including low number of healthy controls, missing data, significant iMO siglec-1 expression in few active patients, and size of harvested monocytes.
Materials and Methods
Lines 319-331: Please explain all abbreviations in Table 1 and/or remove all irrelevant items. The interquartile range (IQR) should be mentioned in this section.
Line 350: Please add a paragraph to explain the detection of Siglec-1 (CD169) expression in different subpopulations. It is important to explain if the expression was normalized to the number and size of the isolated cells.

Author Response
Response to reviewer 2:
We would like to thank the referee very much for the careful review of our manuscript and the constructive criticism. We vigorously revised the manuscript and made changes according to the comments.
- Comment: Abstract Line 14: Please provide the number of IIM patients and controls: … in 62 IIM patients and 10 healthy controls (HC).
Response: The sentence has been completed as requested.
- Comment: Line 15: Please provide “the International Myositis Assessment and Clinical Studies (IMACS)” and remove “62 IIM patients participated in the study.”
Response: The acronym IMACS has been explained. The requested sentence has been deleted.
- Comment: Lines 19-22: Please rephrase the complex sentence.
Response: This part of the abstract has been reworded.
- Comment: Introduction Line 64-66: Please provide references.
Response: The requested reference has been added.
- Comment: Results. Line 108: Please accurately detail the differences between the control and patient groups. According to Table 1, there are obvious differences regarding age and sex. Other characteristics of the control group are not provided in Table 1(NA). Please address the missing data as they may be relevant in terms of inflammatory status. Which other characteristics have been applied to define the control cases?
Response: The reviewer has right, there are differences between control and patient groups, however, these differences were not statistically significant. The distribution of data on age was tested using the Shapiro-Wilk test. Since they did not show a normal distribution, the pooling was terminated with the Mann-Whitney U test. With respect to age, the difference between the healthy controls and the total patient group did not appear to be statistically significant (p=0.098). Fisher’s exact test was used to compare sex ratios and revealed no significant difference (p = 0.178). These results were highlighted in the results section of the modified manuscript. We cannot address any more data of the healthy control group, since they did not have myositis, or any relevant diseases, thus neither disease activity nor manual muscle test were measured. We neither measured the CK, LDH and CRP levels, nor tested the presence of myositis specific/associated antibodies of the healthy individuals as we did not expected any abnormality of these results.
- Comment: Line 123: Figure 1 A shows that only 15-20 patients with IIM express higher Siglec-1 on iMOs. Most of the patients show a lower expression than 200 MFI which is comparable to some controls. In healthy controls, iMOs express significantly higher level of Siglec-1 than noMOs. Please report the results in detail as they appear in the figures.
Response: The reviewer has right. Siglec-1 expression on the different monocyte subsets of myositis patients was very heterogeneous. This part of the result section was modified based on the instructions.
- Comment: Lines 131-132: Please explain the diagrams in detail. Figure legends should explain all indicated characters, cMO, iMO, noMO, etc. In figure 1A, MFI is indicated at the x-axis and should be removed. Please indicate that each grey dot represents the indicated monocyte subset of a single participant.
Response: The referee has totally right. Figure 1 and legend of figure 1 has been modified as requested.
- Comment: Lines 146-147: Please include all abbreviations, applied statistical tests including normality test, and the number of patients in each subgroup into the figure legend. The figure legends should independently provide the reader a sufficient level of information to understand the presented data.
Response: We agree with the reviewer. The proposal has been taken into account for all figure legends and modified accordingly. The legend in Figure 2 has therefore been changed.
- Comment: Line 147: Please revise “. .”.
Response: Thank you. Line 147 has been revised.
- Comment: Lines 167-168: Please follow the comments to figure legends above (Lines 146-147).
Response: We agree with the reviewer. The proposal has been taken into account for all figure signatures and modified accordingly. The legend in Figure 3 has therefore been changed.
- Comment: Lines 195-196 and 202-203: Please provide the Siglec-1 level of cMOs and nomos before and after treatment.
Response: The Siglec levels of the other monocyte subsets were added to the text: “Similarly, the Siglec-1 expression decreased from 299 MFI; 22% to 89 MFI; 0.3% measured on cMOs and from 762 MFI; 29% to 77 MFI; 0.2% on ncMOs..”
- Comment: Figure 4C: The images before and after treatment were obtained from different hands and by different light intensities. Please include better comparable images and crop the facial area as they are not relevant. I suggest the attached layout.
Response: The reviewer has right again. We modified the pictures as requested.
- Comment: Discussion. The authors are kindly advised to reveal the limitations of their study, including low number of healthy controls, missing data, significant iMO siglec-1 expression in few active patients, and size of harvested monocytes.
Response: We agree with the reviewer and as requested, we have highlighted the limitations of the study by inserting a separate paragraph in the discussion section.
- Comment: Materials and Methods Lines 319-331: Please explain all abbreviations in Table 1 and/or remove all irrelevant items. The interquartile range (IQR) should be mentioned in this section.
Response: The abbreviations were explained in the Abbreviations section and the statistical analysis paragraph was modified including the normality distribution and the interquartile range.
- Comment: Line 350: Please add a paragraph to explain the detection of Siglec-1 (CD169) expression in different subpopulations. It is important to explain if the expression was normalized to the number and size of the isolated cells.
Response: The reviewer has right. The flow cytometry section has been expanded and partially reworded as proposed. Modified section was inserted.
Round 2
Reviewer 2 Report
Comments and Suggestions for Authors
I acknowledge the adequate revision of the manuscript and agree with the changes made to a manuscript.